# Calpain and Cardiometabolic Diseases

**DOI:** 10.3390/ijms242316782

**Published:** 2023-11-26

**Authors:** Takuro Miyazaki

**Affiliations:** Department of Biochemistry, Showa University School of Medicine, Tokyo 142-8555, Japan; taku@pharm.showa-u.ac.jp

**Keywords:** alternative mRNA splicing, amino acids, inflammation, dyslipidemia, macrophages, nonalcoholic fatty liver disease (NAFLD), non-alcoholic steatohepatitis (NASH), regulatory T cells, vascular endothelial cells

## Abstract

Calpain is defined as a member of the superfamily of cysteine proteases possessing the CysPC motif within the gene. Calpain-1 and -2, which are categorized as conventional isozymes, execute limited proteolysis in a calcium-dependent fashion. Accordingly, the calpain system participates in physiological and pathological phenomena, including cell migration, apoptosis, and synaptic plasticity. Recent investigations have unveiled the contributions of both conventional and unconventional calpains to the pathogenesis of cardiometabolic disorders. In the context of atherosclerosis, overactivation of conventional calpain attenuates the barrier function of vascular endothelial cells and decreases the immunosuppressive effects attributed to lymphatic endothelial cells. In addition, calpain-6 induces aberrant mRNA splicing in macrophages, conferring atheroprone properties. In terms of diabetes, polymorphisms of the calpain-10 gene can modify insulin secretion and glucose disposal. Moreover, conventional calpain reportedly participates in amino acid production from vascular endothelial cells to induce alteration of amino acid composition in the liver microenvironment, thereby facilitating steatohepatitis. Such multifaceted functionality of calpain underscores its potential as a promising candidate for pharmaceutical targets for the treatment of cardiometabolic diseases. Consequently, the present review highlights the pivotal role of calpains in the complications of cardiometabolic diseases and embarks upon a characterization of calpains as molecular targets.

## 1. Introduction

Obesity, diabetes, hypertension, and dyslipidemia are acknowledged to intersect via systemic metabolism, constituting potent common risk factors. This cluster of maladies is denoted as the metabolic syndrome, with its pernicious cycle of metabolic aberrations culminating in cardiovascular fatality [1,2]. In terms of diabetes, complications encompass chronic kidney diseases, significantly augmenting the propensity for coronary events [3,4]. Furthermore, dyslipidemia, as well as hyperglycemia and obesity, reportedly gives rise to intricacies involving steatohepatitis [5,6]. In contemporary times, this disease area has persistently broadened as a cardiometabolic disorder, prompting numerous epidemiological and molecular biomedical inquiries aimed at its prevention and amelioration. As a result of earlier studies, it was evidenced that statins have remarkable efficacy in rectifying dyslipidemia, with concomitant vasoprotective attributes [7,8]. Incretins and sodium glucose cotransporter 2 (sGLUT2) inhibitors are also ubiquitously employed in the realm of diabetes management, the former manifesting cardioprotective benefits as well as hepatoprotective effects [9,10], while the latter exhibit auspicious nephroprotective properties [11,12]. Hence, therapies endowed with pleiotropic and organoprotective attributes hold promise. Calpain is a stress-responsive intracellular protease, and its involvement in a variety of pathological conditions has been pointed out mainly through basic research since the early 2000s. This family of enzymes has been demonstrated to impart to atherosclerosis, as well as diabetes and hepatic affliction, thereby rendering it an exemplary molecular objective. A lot of chemical inhibitors targeting conventional calpain have been developed, and pharmaceutical companies have undertaken clinical investigations for diseases including Alzheimer’s disease and multiple sclerosis [13]. Given this context, calpain inhibitors have seemed clinically advantageous, and drug repositioning for cardiometabolic diseases may be considered promising. The therapeutic application of calpain inhibitors in cardiometabolic diseases is relatively behind, while recent basic research has elucidated the multifaceted actions of calpain in those metabolic disorders, indicating considerable anticipation for clinical applications. Accordingly, I shall elucidate the pivotal role of calpains within the intricate network of cardiovascular metabolic aberrations and embark upon a scholarly discourse concerning calpains as molecular targets.

## 2. Molecular Basis of Cardiometabolic Calpain Isozymes and Calpastatin

### 2.1. Conventional Calpains and Calpastatin

Calpain designates a superfamily of cysteine proteases characterized by the presence of a CysPC motif within the gene (Figure 1). In mammals, there exist fifteen homologs of the catalytic subunits of calpains, each demonstrating diverse distributions and physiological functions in vivo [14,15,16,17,18]. Among these calpain isozymes, calpain-1 and calpain-2, categorized as conventional isozymes, have garnered global recognition since the 1970s [19,20]. Calpain-1 and calpain-2 form heterodimers by each binding with the common catalytic subunit, calpain-s1. They have undergone comprehensive examination using various methodologies across the fields of enzymology, physiology, and molecular pathology. Conventional calpains exhibit a notable sensitivity to calcium, a characteristic from which their nomenclature, “calpain”, is derived from “calcium” and “papain”. Their activity, however, is subject to regulation by extracellular stresses, including hypoxia, mechanical stresses, inflammatory cytokines, and growth factors, all mediated through the modulation of intracellular calcium levels [21]. In contrast, conventional calpains do not exhibit a strict recognition of a single amino acid sequence, and their substrate specificity remains relatively modest [22]. Conversely, the cleavage sites within each substrate exhibit precise delineation, frequently resulting in discrete fragments discernible as singular bands when subjected to Western blot analysis [23,24]. These substrate fragments may demonstrate divergent biological activities and stabilities compared to their full-length counterparts, thereby potentially influencing cellular functions. The distinct attributes of conventional calpains render them candidates for investigation in numerous disease etiologies. Nevertheless, unequivocally understanding their physiological functions is challenging due to the wide array of drivers and substrates. 

Calpastatin is a specific endogenous calpain inhibitor and is frequently colocalized with calpain proteases [23,25,26]. Structurally, calpastatin comprises a tandem repeat of inhibitory domains that directly interact with and inhibit calpain activity. It is established that the individual inhibitory domains exhibit significant inhibitory efficacy, adequate for the suppression of calpain activity [25]. Dysregulation of calpain-calpastatin balance has been associated with several pathological conditions, such as neurodegenerative diseases (e.g., Alzheimer’s disease and Huntington’s disease) [27,28,29,30], tumor angiogenesis [23], and retinopathy [23]. In these contexts, aberrant calpastatin expression or function can lead to increased calpain activity, contributing to disease progression.

### 2.2. Calpain-10

Several unconventional calpains are also known to contribute to cardiometabolic diseases. Calpain-10 is expressed in various tissues, including pancreatic β-cells [31]. It is comprised of a calpain-type β-sandwich domain and microtubule-interacting and transport domain, as well as CysPC protease domains, while a penta-EF-hand domain is lacking [20]. It was documented that calpain-10 is associated with several diseases, most notably type 2 diabetes mellitus. Its involvement in type 2 diabetes mellitus is linked to genetic variations in the *CAPN10* gene, which encodes calpain-10 [32,33,34].

### 2.3. Calpain-6

Calpain-6 is categorized as an unconventional calpain devoid of proteolytic activity due to the substitution of the cysteine residue within the active site of the CysPC domain with lysine. Its molecular configuration also diverges from conventional calpains, lacking the penta-EF-hand domain responsible for binding to the regulatory subunit [19,20]. Moreover, it has not been documented in a heterodimeric form. The expression profile of calpain-6 exhibits extraordinary distinctiveness. Experimental investigations in murine models have revealed embryonic expression within skeletal and cardiac muscle tissue [35]. Nonetheless, postnatally, this expression is anticipated to wane, with persistent expression primarily confined to the placenta [36]. Notably, *Capn6* (calpain-6 gene)-deficient mice exhibit heightened skeletal muscle capacity and expedited repair of damaged skeletal muscle [36], suggesting potential functional implications of calpain-6 in skeletal muscle development.

## 3. Calpain and Atherosclerosis

Calpains are involved in various comorbidities in cardiometabolic disease, including atherosclerosis (Figure 2). Atherosclerosis represents a direct risk for coronary artery disease and serves as a hallmark of vascular senescence [37]. The pathological signature of atherosclerotic plaques manifests as intimal thickening, concomitant with the accrual of cholesterol, calcification, and necrotic cell cores as the lesion advances [38]. In particular, the formation of the cholesterol-rich necrotic core is attributed to the conversion of macrophages into foamy cells within the vascular intima, triggered by hypercholesterolemia, ultimately culminating in cellular apoptosis [39]. The abundance of macrophages and cholesterol within the lesion is deemed an indicator of plaque stability, with unstable plaques prone to detachment and ensuing embolic events, precipitating coronary artery disease. While these classical principles still command endorsement within the field of atherosclerosis, it has become apparent that research efforts over the past decade have shifted toward investigating inflammation and the immune microenvironment. Notably, the success of the Canakinumab Anti-inflammatory Thrombosis Outcome Study (CANTOS), highlighting canakinumab, an anti-interleukin (IL)-1β antibody, has given credence to a hypothesis that has attracted attention since the beginning of the millennium [40], suggesting atherosclerosis as a form of vasculitis variation. It was reported that both conventional and unconventional calpains contribute to atherosclerotic diseases. 

### 3.1. Conventional Calpain in Vascular Endothelial Cells

Atherosclerosis arises from the infiltration of macrophages into the intimal layer of blood vessels. These cells, if originating from blood monocytes, must traverse the barrier imposed by vascular endothelial cells residing in the intimal superficial stratum. Vascular endothelial cells adopt an epithelial-like monolayer architecture, featuring three principal modalities of intercellular junctions: (1) adherens junctions, (2) tight junctions, and (3) gap junctions [41]. In particular, adherence junctions and tight junctions are reportedly associated with the epithelial barrier functions. The former ostensibly contributes primarily to barrier establishment in peripheral blood vessels, whereas the latter predominates in the blood-brain barrier [41]. Adherence junctions comprise homophilic binding of vascular endothelial (VE)-cadherins within adjacent cells [42,43]. While the cadherin family exhibits various subtypes contingent upon the cell type expressing them, at the inception of our atherosclerosis investigations, at least epithelial (E)-cadherin and neural (N)-cadherin were reported to conventionally display susceptibility to calpain [44,45]. Consequently, we formulated the operative hypothesis that calpain cleaves VE-cadherin in peripheral blood vessels, eliciting endothelial barrier disruption. When subjecting the recombinant VE-cadherin protein to enzymatic proteolysis, we observed the presence of fragments at 95 kDa and 30 kDa, which were located immediately below the intact full-length VE-cadherin molecule with a molecular weight of 125 kDa [46]. This 95 kDa fragment was also discernible within the aorta of *Ldlr*-deficient mice, amplifying with the onset of hypercholesterolemia but exhibiting attenuation upon calpain inhibitor administration. Epitope mapping of VE-cadherin localized the cleavage site proximal to the membrane. The transfection of a calpain-resistant mutant of VE-cadherin into vascular endothelial cells prevented the disruption of adherens junctions or impairment of endothelial barrier function detected in calpain-sensitive mutants. These outcomes suggest that conventional calpain may disrupt the endothelial barrier through paracellular cleavage of VE-cadherin, thereby destabilizing adherens junctions and precipitating a diminution in endothelial barrier function. The cleavage and degradation of VE-cadherin by calpain have been reproduced by several research groups [47,48,49,50]. For example, calpain cleaves VE-cadherin in pulmonary vascular endothelial cells, which appears to cause increased pulmonary vascular permeability [48] and iron-dependent cell death (ferroptosis) [49] in endothelial cells. Calpain-dependent cleavage of VE-cadherin may also be detected in vascular endothelial cells upon transforming growth factor-β1 (TGF-β1) stimulation, which seems to contribute to endothelial-to-mesenchymal transition [50]. Thus, calpain-induced VE-cadherin cleavage has been found to define the function of vascular endothelial cells in various pathophysiological aspects.

In the aforementioned context, vascular endothelial calpain is conspicuously evident within a pathological milieu, precipitating diverse vascular pathologies. However, in the case of atherosclerosis, the etiological factors that drive conventional calpains remain enigmatic. Among the foremost causes of atherosclerosis lies hypercholesterolemia, a condition that extends beyond the elevation of low-density lipoprotein (LDL) levels, encompassing the compositional intricacies of phospholipids within lipoproteins [51,52]. When subjected to denaturing factors, such as oxidative stress or the enzymatic actions of phospholipases, LDL undergoes transformation, leading to the removal of acyl groups from phospholipids, culminating in the formation of lysophospholipids [53]. Given the established activity of certain lysophospholipids within vascular endothelial cells [54,55], which includes calcium transients, it is suspected that oxidative stress-induced and secretory phospholipase A2-induced modifications of LDL can evoke activation of conventional calpain in vascular endothelial cells. Upon subjecting cultured vascular endothelial cells to these modified LDLs, a conspicuous induction of calpain-2 expression was observed, while any alterations in calpain-1 or calpastatin expression were discernibly absent [46]. While the distinctive pattern of expression deviates from the disinhibition of calpastatin observed in pathological angiogenesis, an imbalance in the calpain/calpastatin ratio precipitates in the activation of calpain [23]. Furthermore, no discernible upregulation of calpain-2 occurred within the healthy murine or human aorta; however, its conspicuous manifestation within vascular endothelial cells was evident in severely atherosclerotic lesions. Several basic studies [56,57], including our own [46], have documented that calpain inhibitors efficaciously curtail the progression of atherosclerosis in animal studies, albeit without ameliorating dyslipidemia. This observation suggests that the atherogenic effect of conventional calpains may be localized to the vessel wall. Additionally, it has been reported that simvastatin, a pharmacological agent utilized in the management of dyslipidemia, exerts a dampening effect on calpain activity [58,59], thereby postulating a plausible mechanism underlying the vasoprotective effects associated with statins.

### 3.2. Conventional Calpain in Lymphatic Endothelial Cells

Lymphatic vessels exert a pivotal role in the regulation of fluid homeostasis, the conveyance of immune cells, and the transportation of lipids absorbed from dietary sources [60]. Recent studies have revealed that dysfunctional lymphatic vessels contribute significantly to the pathogenesis and progression of atherosclerosis within major arterial conduits. Notably, lymphatic dysfunction exacerbates dyslipidemia through its propensity to diminish the conveyance of cholesterol from the periphery to the hepatic locus, a phenomenon recognized as reverse cholesterol transport [61]. Furthermore, earlier investigations have ascertained the existence of micro-lymphatic vessels within the adventitia of atherosclerotic plaques, which participate in reverse cholesterol transport and the withdrawal of lymphocytes from the interior plaques [62,63,64]. As lysophospholipids can potentiate conventional calpains, we comprehensively assessed phospholipid composition in a murine model of dyslipidemia by using lipidomic analysis, indicating an augmentation in certain lysophospholipids, including lysophosphatidic acid, within the lymphatic microenvironment [65]. Upon the introduction of these lysophospholipids into cultured lymphatic endothelial cells, the activation of the conventional calpains was ascertained. Furthermore, when lymphocytes isolated from the murine spleen were co-cultured with lymphatic endothelial cells, leading to an evaluation of lymphocyte stability, it was observed that regulatory T cells were stabilized when lysophosphatidic acid was introduced concomitantly with the downregulation of calpain [65]. Upon assessing the expression of factors responsible for stabilizing regulatory T cells within lymphatic endothelial cells, a substantial elevation in TGF-β1 expression was evident, which was further augmented by the concurrent addition of lysophosphatidic acid and the inhibition of calpain. This is attributed to the calpain-induced proteolytic degradation of mitogen-activated protein kinase kinase kinase 1 (MEKK1), a constituent of the TGF-β1 biosynthesis pathway. The supplementation of a TGF-β1 receptor inhibitor in the aforementioned co-culture system canceled the protective influence on regulatory T cells, underscoring the attribution of TGF-β1 derived from lymphatic endothelial cells to this protective effect [65]. Targeting conventional calpains, specifically in lymphatic endothelial cells and myeloid cells, in a murine model of dyslipidemia induced by a high-cholesterol diet, resulted in the suppression of atherosclerosis in the aorta and an elevation in circulating levels of TGF-β1 and regulatory T cell counts [65]. Conversely, lipoprotein cholesterol remained unaffected, suggesting minimal involvement in the reverse cholesterol transport. Pathophysiologic analysis of atherosclerotic lesions revealed a substantial reduction in pro-inflammatory M1-type macrophages due to calpain deficiency, alongside increased expression of immunosuppressive cytokines such as IL-10 and TGF-β1 [65]. These observations predominantly owed to conventional calpains in lymphatic endothelial cells, as they were not replicated in bone marrow chimeric mice solely deficient in calpain in bone marrow cells. In contrast, the administration of TGF-β1 receptor inhibitors nullified the anti-atherosclerotic and regulatory T cell-stabilizing effects observed in lymphatic endothelial cell/myeloid cell-specific calpain-deficient mice, strongly implicating TGF-β1 signaling in these phenomena.

The lymphocyte migration, originating from peripheral organs towards their respective lymph nodes, exhibited a discernible reduction concomitant with the onset of dyslipidemia [60]. Such lymphatic abnormality was subsequently ameliorated through the deficiency of conventional calpains, specifically within the lymphatic endothelial cells or myeloid cells [65]. An RNA-seq analysis within lymph nodes unveiled a noticeable decline in the expression levels of the proinflammatory cytokine IL-18 and vascular cell adhesion molecule-1 (VCAM1). These molecules are highly expressed in lymphatic endothelial cells, both in a murine model of dyslipidemia and in human subjects afflicted with coronary artery disease. In instances where co-cultures of lymphatic endothelial cells and lymphocytes were induced with chemotaxis through chemokine (C-C motif) ligand 19 (CCL19), the rate of lymphocyte migration was substantially inhibited by the simultaneous introduction of lysophosphatidic acid and IL-18 [65]. Conversely, this inhibition was alleviated upon the targeted downregulation of calpain within the lymphatic endothelial cells. The impediment in lymphocyte migration was similarly abrogated through the application of a neutralizing antibody targeting VCAM1. Notably, this effect did not exhibit an additive or synergistic interaction with the downregulation of calpain, suggesting the implication of VCAM1 as one of the primary targets of calpain. These data elucidate the aberrant functioning of calpain within lymphatic endothelial cells in the context of dyslipidemia and its consequent antagonistic impact on the stabilization of regulatory T cells. This, in turn, culminates in a reduction of immunosuppressive effects, particularly within arterial contexts. 

### 3.3. Calpain-6 in Macrophages

As noted above, calpain-6 is localized in skeletal muscle, cardiomyocytes in the fetus, and the placenta [35,36]. In addition, this molecule is localized in osteoblasts when they are differentiated from bone marrow cells in the presence of macrophage colony-stimulating factor and receptor activator of nuclear factor κB ligand [66]. Furthermore, there have been reports indicating that subjecting cancer stem cells to a hypoxic stimulus induces calpain-6 through pluripotency factors Octamer binding factor4 (Oct-4), Nanog homeobox (Nanog), and SRY-box transcription factor 2 (Sox2), potentially mitigating the progression of cellular senescence [67]. Our previous investigations conducted in murine models of atherosclerosis and atherosclerotic lesions in humans have unveiled the robust expression of calpain-6 in foamy macrophages while remaining undetectable in comparatively milder lesions [68]. Within the murine atherosclerosis model, calpain-6 did not exhibit expression in monocytes immediately following their mobilization into the atherosclerotic lesion; rather, its induction occurred within the lesion itself. Therefore, calpain-6 can be functional in the macrophage lineage when they are sufficiently differentiated.

Upon the induction of dyslipidemia in *Capn6*/*Ldlr*-double knockout mice via a high-cholesterol diet, a notable suppression in atherosclerosis formation was observed in comparison to the *Ldlr*-single knockout counterparts [68]. Analysis of bone marrow chimeric mice, selectively targeting *Capn6* in bone marrow cells, faithfully reproduced a phenotype that was observed in the context of systemic deficiency murine models [68]. This emphasizes the functional role of calpain-6 expressed in macrophages. Given the absence of proteolytic functionality in calpain-6, envisioning a scenario in which calpain-6 catalyzes the processing of bioactive proteins, in a manner similar to conventional calpains, remains elusive. In a previous report, calpain-6 was documented to exert microtubule stabilization [69]. Indeed, calpain-6 is associated with guanine nucleotide exchange factor H1 (GEF-H1), a key regulator of RhoGTPase, and colocalizes with microtubules to support the integrity of the cytoskeletal framework. Conversely, the attenuation of calpain-6 seems to destabilize microtubules, liberating GEF-H1 into the cytoplasm, thereby heightening cellular motility. Moreover, the protein expression of Ras-related C3 botulinum toxin substrate 1(Rac1) in bone marrow-derived macrophages exhibited a negative correlation with that of calpain-6 [68]. This downregulation of Rac1 appeared inconsistent with the mRNA expression of *Rac1* confirmed through DNA arrays and quantitative PCR, implying a more intricate regulatory mechanism beyond transcriptional control. A proteomic analysis of calpain-6 immunoprecipitants postulates that calpain-6 interacts with regulatory factors distinct from transcriptional regulators [68]. Among the numerous binding proteins identified, a splicing factor known as “complexed with CEF1 protein 22” (CWC22) was identified, since the splicing factor might exert influence on protein expression without modulating mRNA quantity. CWC22 is recognized as a transport protein that binds to eukaryotic initiation factor 4AIII (eIF4AIII), a constituent of the exon junction complex and a DEAD-box helicase, during the nascent stages of mRNA splicing [70,71]. EIF4AIII induces conformational alterations in mRNA, thus facilitating mRNA processing via the spliceosome by inducing structural modifications to mRNA. Furthermore, studies have affirmed that the downregulation of CWC22 significantly impairs splicing efficiency [72]. In bone marrow-derived macrophages, the Rac1 protein, which was restored by *Capn6* deficiency, underwent downregulation upon further depletion of CWC22 [68]. Notably, in addition to the full-length variant (referred to as isoform1 in the National Center for Biotechnology Information database), a splice variant lacking the fourth exon (termed isoform2 in the same database) has been reported in mouse Rac1. Stimulation with tumor necrosis factor-α (TNF-α) in bone marrow-derived macrophages led to a diminished ratio of isoform1 to isoform 2 mRNA expression, but *Capn6* deletion normalized this ratio, suggesting that CWC22 plays a role in modulating Rac1 splicing.

It was also identified that calpain-6 is implicated in macrophage cholesterol metabolism, often characterized as the central doctrine in atherosclerosis. While class A scavenger receptors and cluster of differentiation 36 (CD36) play pivotal roles in LDL cholesterol uptake by macrophages [73], expression analysis has revealed that calpain-6 does not significantly influence the expression of these receptors [68]. Indeed, no discernible distinction surfaced between wild-type and calpain-6-deficient macrophages concerning oxidized LDL uptake in bone marrow-derived macrophages. Conversely, *Capn6* deficiency resulted in diminished cholesterol uptake when cells were exposed to substantial concentrations of native LDL. This phenomenon signifies that calpain-6 contributes to the receptor-independent, nonspecific uptake of native LDL, referred to as “pinocytosis”. This process, categorized as endocytosis, involves cells engulfing adjacent particles concomitantly with water through spontaneous penetration of the cell membrane [74,75]. *Capn6*-deficient macrophages displayed a limited pinocytosis effect, an effect ameliorated by the downregulation of Rac1 [68]. The intracellular trafficking velocity of vesicles generated during pinocytosis was assessed through labeling and was notably decelerated in the absence of calpain-6, indicating the molecule’s involvement in the intracellular trafficking of these vesicles. *Capn6* deficiency led to an upregulation of Rab11, a marker for recycling endosomes [76], while the expression levels of Rab5 (an early endosomal marker) and lysosomal markers were downregulated. Drawing upon the outcomes of the aforementioned LDL uptake experiments, it is inferred that endosomal recycling, or extracellular efflux, becomes enhanced in calpain-6-deficient macrophages, thus preventing intracellular storage of LDL through lysosomes. The pinocytosis activity is detectable within atherosclerotic lesions of *Ldlr*-deficient mice, manifesting as the uptake of fluorescent nanoparticles [77]. However, in vivo, macrophage pinocytosis activity diminishes due to calpain-6 deficiency [68]. While there are currently limited instances where the pinocytosis effect has been explored as a contributor to atherosclerosis, it is conceivable that this phenomenon, which escalates linearly in response to extracellular LDL concentrations, may surpass receptor-dependent uptake of oxidized LDL [74]. The latter saturate at relatively low concentrations (50–100 μg/mL), which does not adequately mirror the characteristics of atherosclerotic lesions with high LDL concentrations [78]. In contrast to receptor-dependent uptake of oxidized LDL, which plateaus at μg/mL order, the pinocytosis effect, which linearly increases in response to extracellular concentration, is better poised to reflect the characteristics of atherosclerotic lesions with elevated LDL concentrations. Furthermore, there have been reports indicating that the actual quantification of oxidized LDL within the biological system falls short of the amount required for foam cell formation in cultured macrophages [79]. Hence, it is worth considering the necessity of analyzing pinocytosis in order to further comprehend the pathophysiology of atherosclerosis. 

## 4. Calpain and Diabetes

Conventional calpains have been reported to be activated in hyperglycemia to blunt endothelial cells, hepatocytes, and pancreatic β cells (Figure 3). In the initial years of the 21st century, polymorphism of the *CAPN10* (calpain-10 gene) was reported to increase the risks of type 2 diabetes and decrease glucose disposal. Indeed, the G allele of the single nucleotide polymorphism (SNP)-43 has shown associations with insulin resistance [80] and the onset of type 2 diabetes in both cross-sectional [32] and prospective studies [81]. While other investigations have failed to establish a correlation between the G allele and type 2 diabetes or insulin resistance [82,83,84], additional SNPs and/or haplotypes within the corresponding locus have been identified to possess greater significance in glucose metabolism [82,85,86]. Furthermore, variations in this genetic locus might govern insulin secretion [87]. In addition, recent studies have identified that hyperglycemia-induced conventional calpains contribute to myocyte injury in the ischemic heart. 

### 4.1. Calpain and Diabetes

Calpain-10, a constituent of the atypical calpains, stands as the inaugural susceptibility gene identified for type 2 diabetes (T2DM) [32]. T2DM is delineated by diminished insulin secretion, which further augments insulin intolerance. This renders genes affecting insulin exudation as prospective susceptibility genes. *CAPN10* has affiliations with both insulin secretion and sensitivity, thereby facilitating glucose transporter 4 (GLUT4) transposition [88]. Indeed, diminished *CAPN10* expression compromises insulin-induced GLUT4 vesicle transposition, actin reconfiguration, and glucose uptake within adipocytes. The suppression of *CAPN10* impedes insulin-triggered glucose uptake in skeletal muscle as well [89]. The utilization of a calpain inhibitor exerts an impact on actin reorganization and insulin exudation within β cells [90,91]. Short-term exposure augments insulin exudation via hastened exocytosis, whereas 48-hour exposure represses glucose-induced insulin exudation [92]. While precise molecular mechanisms of these cytoskeletal regulations remain enigmatic, embryonic fibroblasts derived from *Capn10^−/−^* mice exhibit a discernible alteration in the localization of microtubule-associated protein 1B (MAP1B), predominantly favoring actin filaments over microtubules [93]. It is ascertained that calpain-10 exerts regulatory dominion over the dynamics of actin through the cleavage of MAP1B. In summary, calpain-10 assumes a pivotal role in both insulin secretion and sensitivity, influencing actin reconfiguration during glucose-induced insulin secretion and insulin-triggered glucose uptake. This is attributable to the processing of MAP1 family proteins.

In addition to calpain-10, conventional calpain is reportedly associated with insulin resistance. Soluble insulin receptor (sIR) comprises the ectodomain of the insulin receptor, which has been identified in human plasma, demonstrating a correlative relationship with blood glucose levels [94]. Yuasa et al. investigated the intricate mechanisms governing insulin receptor (IR) cleavage by utilizing an in vitro model with HepG2 liver-derived cells in order to replicate sIR level variations in diabetic patient plasma [95]. They identified that calpain-2, released into the extracellular milieu via exosomes, orchestrates the direct cleavage of the ectodomain of the IRβ subunit (IRβ), subsequently facilitating intramembrane cleavage of IRβ by γ-secretase. This cleavage event can interfere with the transduction of insulin signaling. Conversely, the mitigation of IR cleavage, achieved through the suppression of calpain-2 and γ-secretase, leads to the restoration of IR substrate-1 and Akt, independently of IR. Furthermore, the antihyperglycemic agent metformin mitigates IR cleavage, concomitant with the inhibition of calpain-2 release in exosomes, thereby recovering insulin signaling [95]. Notably, in individuals with type 2 diabetes, plasma sIR levels exhibit an inverse correlation with insulin sensitivity. Recently, it was documented that sIR concentrations exhibited a positive correlation with estrogen levels in pregnant women, demonstrating a significant elevation during the advanced stages of pregnancy, independently of glucose concentrations [96]. In vitro experiments showed estrogen can induce IR cleavage, thereby compromising cellular insulin signaling [96]. The cleavage of IR prompted by estradiol was mitigated through the inhibition of calpain-2 and γ-secretase. Estrogen manifested these physiological impacts via the G protein-coupled estrogen receptor, and its selective ligand augmented the expression of calpain-2 while facilitating the secretion of exosomes, leading to a marked increase in extracellular calpain-2. The concurrent exposure to estrogen and elevated glucose levels exhibited a synergistic influence on IR cleavage. Metformin averted the release of calpain-2 within exosomes and reinstated insulin signaling impaired by estrogen. Thus, calpain-2 has an integral role in the metabolic processes of sIR, thereby actively contributing to insulin resistance both during the peripartum period and in the context of diabetes.

### 4.2. Calpain and Diabetic Cardiomyopathy

Previous research showed that targeted inhibition of calpain within endothelial cells mitigated cardiac fibrosis, hypertrophy, and myocardial dysfunction in type 1 and type 2 diabetic mice while leaving systemic metabolic parameters unaltered. This cardioprotective effect correlated with heightened myocardial capillary density and coronary flow reserve [97]. Ex vivo analysis unveiled augmented neovascularization in *Capns1*-knockout mice, and in cultured endothelial cells, calpain inhibition ameliorated angiogenic tube formation and averted apoptosis by upregulating β-catenin protein levels. These discoveries underscore the pivotal role of calpain in diabetic cardiomyopathy through the suppression of β-catenin. Another series of investigations identified the role of conventional calpains in heightened susceptibility to ischemia-reperfusion injury in diabetic myocardium in mice [98]. Autophagic flux exhibited compromise within ischemia-reperfusion-subjected hearts, particularly in diabetic subjects, manifesting as impaired autophagosome formation and clearance mechanisms. Conventional calpains were extensively activated in diabetic ischemia-reperfusion models, and the inhibition of calpain activity demonstrated improvements in cardiac function, a reduction in cell mortality, and a partial restoration of autophagic flux [98]. Autophagy-related proteins, autophagy-related 5 (Atg5) and lysosome-associated membrane protein 2 (LAMP2), underwent degradation in diabetic ischemia-reperfusion-afflicted myocardium due to calpain activation, whereas their levels were restored upon calpain inhibition. Co-overexpression of Atg5 and LAMP2 ameliorated myocardial injury and normalized autophagic flux [98]. Therefore, hyperglycemia abrogates ischemia-reperfusion-induced autophagic flux, thereby inducing cardiac dysfunction, which is attributed to calpain activation and cleavage of Atg5/LAMP2.

Furthermore, it was noted that increased calpain-1 and -2 levels and activities within cardiac mitochondria have been correlated with conditions such as ischemia, diabetes, and sepsis, thus contributing significantly to myocardial injury [99]. To evaluate the protective role of mitochondrial calpain inhibition, transgenic murine models featuring cardiomyocyte-specific overexpression of mitochondria-targeted calpastatin were generated. These murine subjects exhibited exclusive mitochondrial calpastatin expression within the cardiac tissue and were subsequently subjected to global ischemia/reperfusion injury as well as induced hyperglycemic conditions. The mitochondria-targeted overexpression of calpastatin markedly attenuated oxidative stress and apoptotic processes in isolated hearts during ischemia/reperfusion injury, while simultaneously ameliorating cardiac function and remodeling in hyperglycemic murine models [99]. These protective effects were concomitant with augmented ATP synthase, H^+^ transporting, mitochondrial F1 complex, alpha subunit 1 (ATP5A1) protein expression, and enhanced ATP synthase activity. In cultured myoblasts, the presence of mitochondria-targeted calpastatin served to preserve ATP5A1 levels, diminish mitochondrial reactive oxygen species production, and decrease cell apoptosis during hypoxia/reoxygenation [99]. These findings suggest the potential roles of cytoplasmic and mitochondrial calpains in myocardial injury and aberrant cardiac functions in a diabetic ischemic heart. 

## 5. Calpain and Liver Disease

Non-alcoholic fatty liver disease (NAFLD) encompasses a spectrum ranging from mild hepatic steatosis to non-alcoholic steatohepatitis (NASH), frequently manifesting concomitantly with metabolic disorders, such as diabetes, dyslipidemia, and obesity [5,6]. Cumulative evidence from our research and other studies indicates the involvement of conventional calpains in the pathogenesis of these hepatic conditions (Figure 4).

### 5.1. Conventional Calpains and Amino Acid Metabolism in the Liver

It has long been established that steatotic hepatic disorders are influenced by perturbations in amino acid metabolism. Branched-chain amino acids (BCAAs), which are comprised of leucine, isoleucine, and valine, exhibit therapeutic efficacy in mitigating cachexia and sarcopenia [100,101], albeit concomitant reports of exacerbating type 2 diabetes, obesity, and ischemic heart disease [102,103]. Furthermore, emerging evidence posits that the introduction of the amino acid transporter sodium-coupled neutral amino acid transporter 2 into hepatic tissue exacerbates hepatic steatosis [104], implicating specific amino acids in hepatocellular lipid accumulation. Regarding nutrient deprivation, amino acids are supplied through autophagy-mediated protein catabolism and proteasomal systems [105,106]. Since conventional calpains have been reported to exert limited proteolysis, these proteases are incapable of directly generating amino acids. In a recent study by Matouschek et al., calpain-mediated proteolytic processing has emerged as a mechanism amplifying susceptibility to ubiquitin-associated proteasomal degradation [107]. Based on these observations, we have focused on calpain’s ability to cleave a wide spectrum of target proteins, assessing our hypothesis that this protease may function as the rate-limiting factor in proteolytic amino acid synthesis [108]. When cultured vascular endothelial cells were subjected to hyperglycemic conditions, calpain was activated, and amino acids were extracellularly released in a calpain activity-dependent manner. Metabolomic analysis of the conditioned medium unveiled that at least 16 extracellular amino acids, alongside dipeptides and related metabolites, were contingent on intracellular calpain activity. When the conditioned media derived from calpain-activated endothelial cell culture were applied to the hepatocyte cell line HepG2 and subsequently stimulated with insulin, phosphorylation of S6K and subsequent de novo lipogenesis was observed. Conversely, no such lipogenic response was observed when employing the control media obtained from calpain-inactive endothelial cell culture. Upon analyzing the impact of the mammalian target of rapamycin complex 1 (mTORC1) inhibitor and L-type amino acid transporter 1 (LAT1) inhibitor in the aforementioned culture systems, both drugs significantly inhibited S6K phosphorylation and the development of lipid droplets [108]. Hence, it is thought that amino acids in the conditioned media induce de novo lipogenesis through the mTORC1/S6K axis. Additionally, the deletion of endothelial and myeloid calpain systems, achieved by targeting *Capns1* through the utilization of the Tek promoter, markedly suppressed hepatic triglyceride accumulation [108]. This intervention exhibited no detectable effect on blood triglyceride and cholesterol profiles. Conversely, mice characterized by the exclusive absence of calpain-s1 in bone marrow-derived cells, established via bone marrow transplantation, manifested no discernible impact arising from the aforementioned genetic modification, particularly with regard to fatty liver development. Consequently, it is reasonable to postulate that vascular endothelial cell-derived calpain may exert a significant influence on the pathogenesis of fatty liver. Notably, the quantification of amino acid levels within the hepatic tissue of these mice unveiled a pronounced decrease in leucine, isoleucine, and glycine content in vascular endothelial *Capns1* knockout subjects [108]. Furthermore, a reduction in liver BCAA levels was also observed in these genetically manipulated mice, with no concomitant alteration in blood BCAA levels. An administration of the LAT1 inhibitor to Flox mice subjected to a high-fat diet resulted in a reduction in hepatic triglyceride levels. Such pharmacological effects were indiscernible when the compound was administered to conditional calpain-s1 knockout mice [108]. Conversely, this drug did not exert any perceivable influence on blood triglyceride or amino acid levels. Thus, it is conceivable that vascular endothelial calpains are implicated in the modulation of hepatic amino acid levels, thereby exerting a substantial impact on the pathogenesis of fatty liver.

### 5.2. Conventional Calpains and Hepatocellular Insults

While wild-type mice subjected to the high-fat diet exhibited a notable elevation in serum aspartate aminotransferase (AST) and alanine aminotransferase (ALT) levels, the *Capns1* knockout mice exposed to the HFD showed a marked reduction in these markers for hepatic injury [109]. Furthermore, the *Capns1* knockout led to a substantial decrement in oxidized LDL, malondialdehyde, tumor necrosis factor-α, and interleukin-6 levels, coupled with an augmentation in superoxide dismutase activity, without exerting any discernible impact on lipid profiles [109]. Conversely, when maintained on the low-fat diet, those liver and redox parameters remained largely homogenous between the wild-type and *Capns1* knockout mice, with the exception of a decreased calpain enzymatic activity within the hepatic tissues. This suggests that calpain-1 can reduce the integrity of redox systems during hepatic dysfunction induced by a high-fat diet. 

It was also postulated that conventional calpains play a pivotal role in the regulation of fibrogenic extracellular matrix homeostasis during dermal wound healing [110]. In this case, calpains in endothelial cells potentiated fibrogenic responses in adjacent fibroblasts via platelet-derived growth factor signaling. On the other hand, Sato et al. observed a positive correlation between the expression of calpain-2 and the severity of fibrosis in NASH among human subjects [111]. Furthermore, they established a NASH model by intraperitoneally administering carbon tetrachloride to murine subjects and subsequently monitored the circulating proteolytic products through “degradome” analysis until hepatic fibrosis regressed to an undetectable state. The proteomic analysis revealed an accumulation of proteolytic fragments during the recovery phase of liver disorders, which appears to be attributed to fibrotic insults. In silico analysis indicated that the peptide sequence profile matches the enzymatic characteristics of conventional calpains. Interestingly, calpain activity exhibited an upregulation during the recovery phase in carbon tetrachloride-induced NASH mice, although this phenomenon warrants functional validation in future investigations.

4-hydroxynonenal is a lipid peroxidation product formed through the oxidation of polyunsaturated fatty acids, specifically omega-6 fatty acids like linoleic acid, and is commonly used as a marker for oxidative stress and cellular damage [112]. This oxidative adduct induces hepatocellular apoptosis through its disruption of the lysosomal limiting membrane [113]. It is noteworthy that the accumulation of 4-hydroxynonenal within hepatocytes in human subjects suffering NASH displays greater severity in cases characterized by heightened levels of inflammation, ballooning, and fibrosis, and is concurrent with observable lysosomal hypertrophy. Importantly, 4-hydroxynonenal initiates the activation of calpain-1 and calpain-2 through its interaction with the G-protein coupled receptor 120, thereby instigating the demise of the lysosomal membrane, cathepsin leakage, and subsequent hepatocellular apoptosis. Strategies involving the inhibition of G-protein coupled receptor 120 or calpain expression, as well as the application of Alda-1, an aldehyde dehydrogenase 2 agonist, for the degradation of 4-hydroxynonenal, have demonstrated efficacy in mitigating hepatocellular apoptosis [113]. Consistently, the administration of 4-hydroxynonenal to simians and Alda-1 to murine subjects has yielded congruent outcomes, underscoring the implication of 4-hydroxynonenal in hepatocellular mortality and the subsequent proposition of potential therapeutic modalities for the attenuation of NASH-associated hepatic fibrogenesis. 

### 5.3. Conventional Calpains and Hepatic Ischemic Insults

Ischemia-reperfusion injury commonly occurs during hepatic resection and transplantation, particularly presenting challenges in elderly liver cases. Despite the presence of a potent endogenous calpain inhibitor known as calpastatin, ischemia-reperfusion triggers calpain activation, resulting in compromised autophagic processes, mitochondrial dysfunction, and hepatocyte apoptosis [114]. Substantial hepatocellular apoptosis was detectable, particularly in reperfused aged hepatocytes. Intriguingly, the expression of calpastatin was essentially elevated in unstimulated hepatocytes derived from aged mice, while reperfusion elicited a significant reduction in calpastatin levels thereby exacerbating age-related hepatic injury following ischemia-reperfusion. In contrast, calpastatin levels in hepatocytes from young mice were sustained even during reperfusion insults. It seems that the downregulation of calpastatin may be attributed to proteasomal degradation. MicroRNA miR-140-5p has been implicated in the pathogenesis of liver ischemia-reperfusion and hypoxia/reoxygenation of AML12 cells [115]. Overexpression of miR-140-5p reduced liver and cell injury, with miR-140-5p targeting calpain-1 confirmed by dual-luciferase reporter assays. These findings suggest miR-140-5p may serve as a therapeutic target for liver ischemia-reperfusion injury by modulating calpain-1. Thus, conventional calpains are closely related to ischemia-reperfusion injury even in the liver. 

## 6. Calpain and Obesity

Obesity exacerbates atherosclerosis, diabetes, and liver disease through complex biological mechanisms. Indeed, obesity contributes to the progression of atherosclerosis, a condition characterized by the buildup of cholesterol-prone plaques in arterial walls. Excess adipose tissue in obese individuals leads to an increase in circulating free fatty acids and pro-inflammatory cytokines, and a decrease in cardioprotective cytokines, such as adiponectin [116]. Consequently, a chronic inflammatory state and endothelial dysfunction driven by the metabolic abnormalities noted above ensue, fostering the formation of atherosclerotic lesions. Moreover, obesity is a major risk factor for the development of type 2 diabetes mellitus. Excess adipose tissue, particularly in the abdominal region, is associated with increased secretion of pro-inflammatory substances, which exacerbate insulin resistance. In obesity, the increased delivery of free fatty acids to the liver, as well as insulin resistance, leads to an enhanced deposition of fat in the liver [117]. Inflammatory processes, oxidative stress, and the release of pro-fibrotic factors in obese individuals contribute to those complications. There is no evidence that calpain systems potentiate adipose tissue expansion during obesity so far. Indeed, the overexpression of calpastatin had no discernible impact on the body weight and adipose mass gain induced by a high-fat diet [118]. Rather, the inhibition of calpain exhibited a transient enhancement in glucose tolerance during high-fat diet exposure. However, the overexpression of calpastatin significantly mitigated adipocyte apoptosis and adipose tissue collagen deposition. Furthermore, inhibition of conventional calpain markedly attenuated the inflammatory responses elicited by obesity within the adipose tissue and suppressed macrophage migration to the adipose tissue in vitro [118]. Thus, conventional calpains have diverse functions such as apoptosis, fibrosis, and inflammation during the high-fat diet-induced expansion of adipose tissue. 

## 7. Conclusions and Future Perspective

Calpain becomes activated or induced in cardiometabolic diseases, significantly contributing to the progression of associated complications. According to findings from animal studies, it seems that the activation of calpain does not influence metabolic parameters; instead, it primarily influences tissue susceptibility. As mentioned above, conventional calpain activated by cardiometabolic diseases has been found to contribute to functional impairments and cytotoxic effects in at least the vascular, lymphatic, myocardial, and hepatic cell types. Consequently, intervening in the conventional calpain system is anticipated to confer a protective impact across various organs. This characteristic could prove highly advantageous in managing cardiometabolic disease encompassing a broad array of comorbidities. Given the multitude of chemical inhibitors available for conventional calpain, future research should be directed toward applied investigations with an emphasis on pharmacological assessment and clinical application. As indicated above, calpain-6 is observed to stabilize macrophage pinocytosis and worsen atherosclerosis. Notably, phosphoinositide 3-kinase, dynamin, microtubules, actin, and Rho GTPase have all been identified as contributors to the process of LDL uptake through pinocytosis. Furthermore, it has been established that the loss of Na+/H+ exchanger 1 in macrophages actively suppresses the accumulation of native LDL, subsequently impeding the progression of atherosclerosis via pinocytosis within these cellular entities. Such pinocytosis may be effectively prevented through the utilization of the Na+/H+ exchanger 1 inhibitor, specifically, 5-(N-Ethyl-N-isopropyl)amiloride, which offers a viable alternative to the currently employed antidepressant, imipramine. Given that calpain-6 manifests intermolecular interactions with CWC22, this biochemical process may constitute a promising candidate for pharmaceutical intervention. Furthermore, as calpain-6 exerts its influence through mRNA splicing, it becomes imperative to contemplate a specific splice variant as a potential target, and the modulation of cellular functions through oligonucleic acids presents itself as a viable strategy. In order to realize such a clinical strategy, additional investigations are deemed necessary to acquire a more comprehensive body of information pertaining to the variations and specificity concerning the calpain target. Nevertheless, it is challenging to assert that the contributions of unconventional calpains other than calpain-6 and calpain-10 to cardiometabolic diseases have been adequately examined. In recent years, the majority of unconventional calpain-deficient mice have been generated, many of which seem to exhibit a phenotype not characterized by perinatal lethality. It is intriguing that future research utilizes these genetically modified animals to explore the pathophysiological relevance of cardiometabolic diseases.

## Figures and Tables

**Figure 1 ijms-24-16782-f001:**
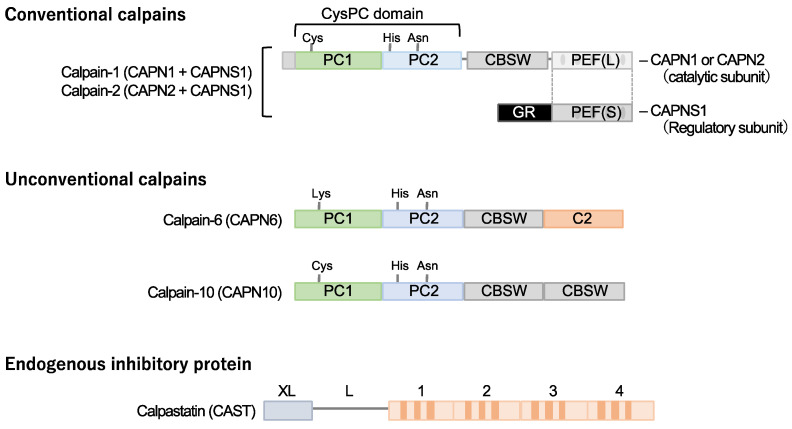
Cardiometabolic calpains. Calpains are a superfamily of cysteine proteases with a CysPc domain. Two types of conventional isozymes and two types of unconventional calpains are thought to be involved in cardiometabolic diseases. Conventional calpains are formed when the subtype-specific catalytic subunits calpain-1 and calpain-2 form heterodimers with the common regulatory subunit, calpain-s1, respectively. On the other hand, no quaternary structure has been identified for calpain-6 and calpain-10, which are unconventional isoforms. Calpastatin is known as a specific endogenous inhibitor, and its increased expression in cells may downregulate the activity of conventional isozyme and calpain-10. Notably, a cysteine residue in the active core of calpain-6 is substituted with lysine and is thought to lack protease activity. CBSW, calpain-type β-sandwich domain; CysPc, cysteine protease domain, calpain-type; GR, glycine-rich domain; PC, protease core; PEF, penta-EF-hand domain.

**Figure 2 ijms-24-16782-f002:**
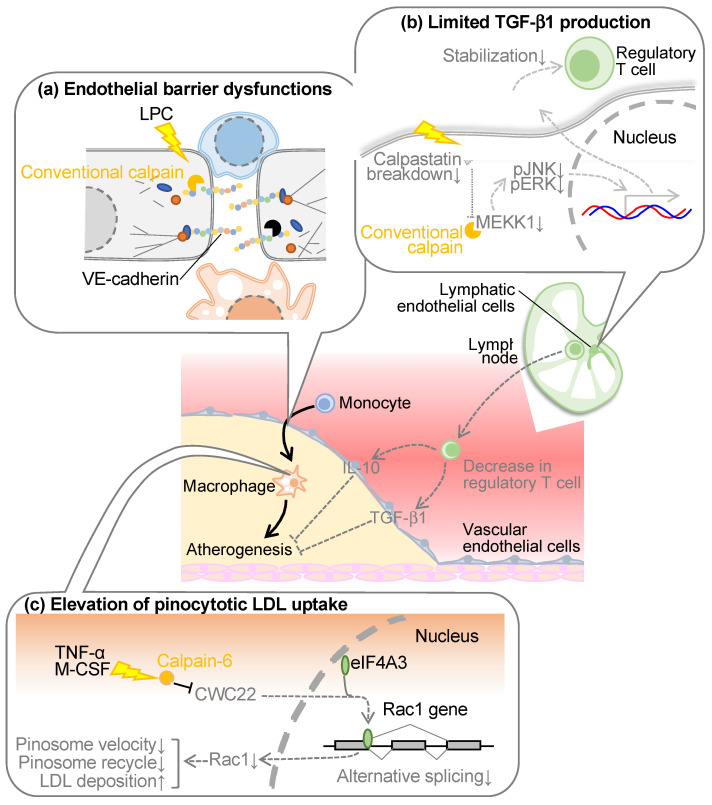
Conventional and unconventional calpains potentiate atherosclerosis. (**a**) The mechanism involves calpain-induced cleavage of VE-cadherin between vascular endothelial cells, which destabilizes the adherence junction and promotes the recruitment of monocytes to the vascular intima. (**b**) Conventional calpain is concomitantly activated in lymphatic endothelial cells in the presence of dyslipidemia, resulting in reduced stability of regulatory T cells via TGF-β1 production and reduced immunosuppression in arteries, contributing to atherosclerosis development. (**c**) Calpain-6 is induced in macrophages in atherosclerotic lesions and promotes pinocytosis-mediated LDL cholesterol accumulation by modifying CWC22-mediated alternative splicing. CWC22: complexed with CEF1 protein 22; eIF4A3: eukaryotic translation initiation factor 4A3; ERK: extracellular signal-regulated kinase; IL-10: interleukin-10; JNK: c-Jun N-terminal kinase; LDL: low-density lipoprotein; LPA: lysophosphatidic acid; LPC: lysophosphatidylcholine; M-CSF: macrophage colony-stimulating factor; MEKK1: mitogen-activated protein kinase kinase kinase 1; Rac1: Ras-related C3 botulinum toxin substrate 1; TGF-β1: transforming growth factor-β1; TNF-α: tumor necrosis factor-α.

**Figure 3 ijms-24-16782-f003:**
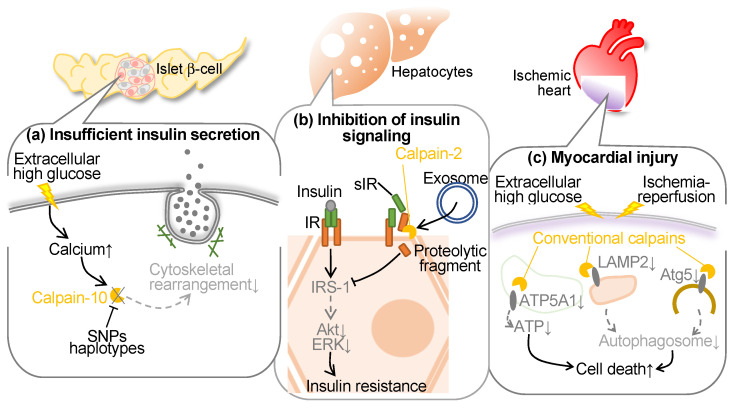
Conventional and unconventional calpains are involved in the pathogenesis and comorbidities of diabetes. (**a**) Calpain-10 is known to contribute to insulin release in pancreatic islet β-cells, but this function is impaired by single nucleotide polymorphisms (SNPs) including SNP-43 and/or haplotype on *CAPN10* gene. (**b**) Extracellular calpain-2, released through exosomes, orchestrates the direct cleavage of the IRβ subunit ectodomain. This cleavage hinders insulin signaling transduction through IRS-1/Akt/ERK. Individuals with type 2 diabetes show an inverse correlation between plasma sIR levels and insulin sensitivity. (**c**) Conventional calpain also contributes to ischemic myocardial injury in diabetes. In this case, mitochondrial calpain degrades ATP5A1 to reduce ATP production and downregulates autophagy through LAMP2 and Atg5 degradation to promote cell death. Atg5: autophagy related 5; ATP: adenosine triphosphate; ATP5A1: ATP synthase, H+ transporting, mitochondrial F1 complex, alpha subunit 1; ERK: extracellular signal-regulated kinase; IR: insulin receptor; IRS-1: insulin receptor substrate-1; LAMP2: lysosome-associated membrane protein 2; sIR: soluble insulin receptor; SNP: single nucleotide polymorphism.

**Figure 4 ijms-24-16782-f004:**
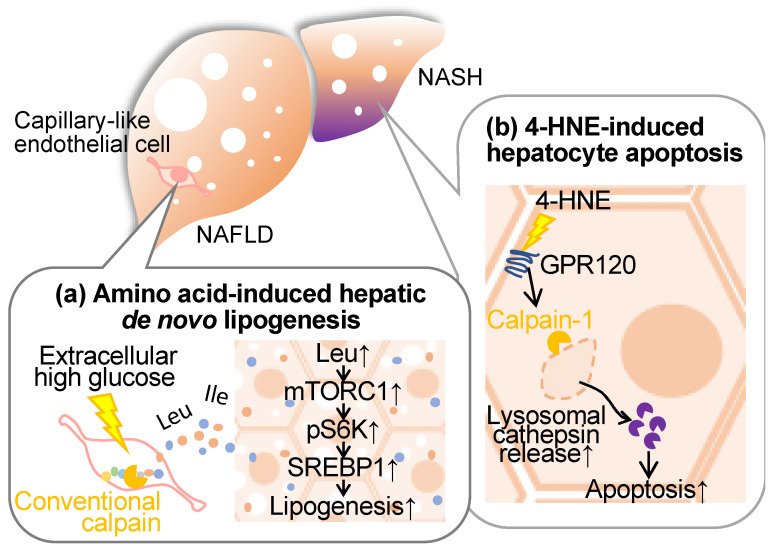
Conventional calpains contribute to liver disease. (**a**) In the capillary-like endothelial cells found in parenchyma in NAFLD liver, conventional calpains are over-activated and execute excessive proteolysis resulting in an increase of amino acids, including leucine, in the liver microenvironment. This increase in amino acids promotes de novo lipogenesis in the surrounding hepatocytes and accelerates the formation of fatty liver. (**b**) the oxidative adduct 4-HNE activates hepatocyte calpain in NASH via GPR120, which permeabilizes the lysosomal membrane and increases cytosolic cathepsins. As a result, hepatocytes are thought to undergo cell death. GPR120: G-protein coupled receptor 120; 4-HNE: 4-Hydroxynonenal; mTORC1: mammalian target of rapamycin complex 1; NAFLD: non-alcoholic fatty liver disease; NASH: non-alcoholic steatohepatitis; SREBP1: sterol regulatory element-binding protein 1; S6K: ribosomal protein S6 kinase B1.

## Data Availability

Data are contained within the article.

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
