# Peer review of "Calpain and Cardiometabolic Diseases"

_ijms, 2023, doi:10.3390/ijms242316782_

Round 1

Reviewer 1 Report

Comments and Suggestions for Authors

The manuscript's structure appears to cover various topics related to calpain and its role in cardiometabolic diseases. It delves into the molecular basis, various isozymes, and their connections to atherosclerosis, diabetes, liver disease, and obesity. The manuscript's structure and coverage of the topic show promise, and with the suggested enhancements in clarity, language, organization, and engagement, it has the potential to be a valuable contribution to the field of calpain's role in cardiometabolic diseases. A well-revised manuscript addressing the mentioned suggestions should significantly improve its quality, making it more accessible and informative to the target audience.

Certainly, here are some potential weaknesses or areas for improvement in the manuscript:

Major comments

1.       The abstract and introduction sections could benefit from simplified language. These sections should provide a clear and engaging overview of the topic to capture the readers' interest. Complex terminology and lengthy sentences may be challenging for some readers.

2.       The abstract should provide a concise summary of the entire manuscript, including the main findings or conclusions. Currently, it focuses mostly on introducing the topic and its importance.

3.       Ensure that all claims are properly supported by relevant citations. Citations and references should follow a consistent and accurate format.

4.       The introduction should engage the reader by clearly stating the purpose of the review and what the reader can expect to learn or gain from reading the manuscript. Please revise this section accordingly.

5.       Consider the balance of content across sections. Some sections might require more in-depth coverage, while others could be summarized more concisely.

6.       The "Conclusions and future perspective" section should summarize key findings and their implications. It's also essential to provide a clear call to action or suggestions for future research. Please revise this section accordingly.

Comments on the Quality of English Language

In summary, while the manuscript shows promise with its comprehensive structure, it would benefit from improvements in clarity, language, and engagement in the abstract and introduction sections. Additionally, ensuring thorough citations and references throughout the manuscript is vital. Overall, with these suggested enhancements, the manuscript can become a valuable contribution to the understanding of calpain's role in cardiometabolic diseases.

Author Response

[Author] Thank you so much for your valuable comments and suggestions. Here I will address your question and suggestion in a point-to-point manner.

The manuscript's structure appears to cover various topics related to calpain and its role in cardiometabolic diseases. It delves into the molecular basis, various isozymes, and their connections to atherosclerosis, diabetes, liver disease, and obesity. The manuscript's structure and coverage of the topic show promise, and with the suggested enhancements in clarity, language, organization, and engagement, it has the potential to be a valuable contribution to the field of calpain's role in cardiometabolic diseases. A well-revised manuscript addressing the mentioned suggestions should significantly improve its quality, making it more accessible and informative to the target audience.

 Certainly, here are some potential weaknesses or areas for improvement in the manuscript:

Major comments

  1. The abstract and introduction sections could benefit from simplified language. These sections should provide a clear and engaging overview of the topic to capture the readers' interest. Complex terminology and lengthy sentences may be challenging for some readers.
  2. The abstract should provide a concise summary of the entire manuscript, including the main findings or conclusions. Currently, it focuses mostly on introducing the topic and its importance.

[Author] According to comments 1. and 2. the abstract was rewritten. Specifically, some words were replaced to improve readability. Some instances of calpain-associated cardiometabolic complications were noted.

–––– Calpain is defined as a member of the superfamily of cysteine proteases possessing the CysPC mo-tif within the gene. Calpain-1 and -2, which are categorized as conventional isozymes, execute limited proteolysis in a calcium-dependent fashion. Accordingly, the calpain system participates in physiological and pathological phenomena, including cell migration, apoptosis, and synaptic plasticity. Recent investigations have unveiled the contributions of both conventional and uncon-ventional calpains to the pathogenesis of cardiometabolic disorders. In the context of atheroscle-rosis, overactivation of conventional calpain attenuates the barrier function of vascular endothe-lial cells and decreases the immunosuppressive effects attributed to lymphatic endothelial cells. In addition, calpain-6 induces aberrant mRNA splicing in macrophages, conferring atheroprone properties. In terms of diabetes, polymorphisms of the calpain-10 gene can modify insulin secre-tion and glucose disposal. Moreover, conventional calpain reportedly participates in amino acid production from vascular endothelial cells to induce alteration of amino acid composition in the liver microenvironment thereby facilitating steatohepatitis. Such multifaceted functionality of calpain underscores its potential as a promising candidate for pharmaceutical targets for the treatment of cardiometabolic diseases. Consequently, the present review highlights the pivotal role of calpains in the complications of cardiometabolic diseases and embarks upon a character-ization of calpains as molecular targets. (lines 6-22 in the revised manuscript)

  1. Ensure that all claims are properly supported by relevant citations. Citations and references should follow a consistent and accurate format.

[Author] The citations and their list were confirmed and corrected appropriately. I've tried to cite frequently and added some references.

  1. The introduction should engage the reader by clearly stating the purpose of the review and what the reader can expect to learn or gain from reading the manuscript. Please revise this section accordingly.

[Author] The purpose of the review was modified as below.

–––– Calpain is a stress-responsive intracellular protease, and its involvement in a variety of pathological conditions has been pointed out mainly through basic research since the early 2000s. This family of enzymes has been demonstrated to impart to atherosclerosis, as well as diabetes and hepatic affliction, thereby rendering it an exemplary molecular objective. A lot of chemical inhibitors targeting conventional calpain have been developed, and pharmaceutical companies have undertaken clinical investigations for diseases including Alzheimer's disease and multiple sclerosis [13]. Given this context, calpain inhibitors have seemed clinically advantageous, and drug repositioning for cardiometabolic diseases may be considered promising. The therapeutic application of calpain inhibitors in cardiometabolic diseases is relatively behind, while recent basic research has elucidated the multifaceted actions of calpain in those metabolic disorders, indicating considerable anticipation for clinical applications. Accordingly, I shall elucidate the pivotal role of calpains within the intricate network of cardiovascular metabolic aberrations, and embark upon a scholarly discourse concerning calpains as molecular targets. (lines 41-55 in the revised manuscript)

  1. Consider the balance of content across sections. Some sections might require more in-depth coverage, while others could be summarized more concisely.

[Author] The topic “calpain and diabetes” was expanded. Accordingly, Figure 3 and its legends were modified.

  1. The "Conclusions and future perspective" section should summarize key findings and their implications. It's also essential to provide a clear call to action or suggestions for future research. Please revise this section accordingly.

[Author] “Conclusions and future perspective” was expanded to elucidate my proposition more clearly.

––––Calpain becomes activated or induced in cardiometabolic diseases, significantly contributing to the progression of associated complications. According to findings from animal studies, it seems that the activation of calpain does not influence metabolic parameters; instead, it primarily influences tissue susceptibility. As mentioned above, conventional calpain activated by cardiometabolic diseases has been found to contribute to functional impairments and cytotoxic effects in at least the vascular, lymphatic, myocardial, and hepatic cell types. Consequently, intervening in the conventional calpain system is anticipated to confer a protective impact across various organs. This characteristic could prove highly advantageous in managing cardiometabolic disease encompassing a broad array of comorbidities. Given the multitude of chemical inhibitors available for conventional calpain, future research should be directed toward applied investigations with an emphasis on pharmacological assessment and clinical application. As indicated above, calpain-6 is observed to stabilize macrophage pinocytosis and worsen atherosclerosis. Notably, phosphoinositide 3-kinase, dynamin, microtubules, actin, and Rho GTPase have all been identified as contributors to the process of LDL uptake through pinocytosis. Furthermore, it has been established that the loss of Na+/H+ exchanger 1 in macrophages actively suppresses the accumulation of native LDL, subsequently impeding the progression of atherosclerosis via pinocytosis within these cellular entities. Such pinocytosis may be effectively prevented through the utilization of the Na+/H+ exchanger 1 inhibitor, specifically, 5-(N-Ethyl-N-isopropyl)amiloride, which offers a viable alternative to the currently employed antidepressant, imipramine. Given that calpain-6 manifests intermolecular interactions with CWC22, this biochemical process may constitute a promising candidate for pharmaceutical intervention. Furthermore, as calpain-6 exerts its influence through mRNA splicing, it becomes imperative to contemplate a specific splice variant as a potential target, and the modulation of cellular functions through oligonucleic acids presents itself as a viable strategy. In order to realize such a clinical strategy, additional investigations are deemed necessary to acquire a more comprehensive body of information pertaining to the variations and specificity concerning the calpain target. Nevertheless, it is challenging to assert that the contributions of unconventional calpains other than calpain-6 and calpain-10 to cardiometabolic diseases have been adequately examined. In recent years, the majority of unconventional calpain-deficient mice have been generated, many of which seem to exhibit a phenotype not characterized by perinatal lethality. It is intriguing that future research utilizes these genetically modified animals to explore the pathophysiological relevance of cardiometabolic diseases. (lines 645-678 in the revised manuscript)

Reviewer 2 Report

Comments and Suggestions for Authors

The article titled “Calpain and cardiometabolic diseases” was reviewed. 

The aim of the review was to point out the role of calpains in the complications of cardiometabolic diseases and molecular targets. 

The author starts briefly describing different calpains in terms of their structure and homology. Subsequently, there is information over the molecular basis of cardiometabolic calpain isozymes and calpastatin. Finally, the author structured the manuscript in different sections over calpain and atherosclerosis, diabetes, liver disease and obesity. 

The author has a broad number of publications on Calpain function, regulation, and isoforms. Which they are mostly presented in the review.  

Comments and suggestions 

It is clearly understood this is not a Systematic review. However, it is suggested to give an overview of the researching process to gather and filtering the information. If it Is the purpose to present all the information since its discovery and relation with cardiometabolic traits or any criteria to include the articles (outcomes, methodology such as in vivo and in vitro models, etc.).

It should be specified the calpains included in the manuscript are 1, 2, 6 and 10 because of their involvement in the cardiometabolic traits and consequences, and they are presented as conventional and unconventional calpains. 

Please check the connector “and” in the following idea, lines 292-294: 

In bone marrow derived macrophages, the expression of RhoGTPase and ascertained that the protein expression of Rac1 displayed an apparent negative correlation with calpain-6.

The figures (graphical material) give a concise message together with the written information. In this sense, it is necessary to structure the figures to clarify the idea. In figure legend 2, it is explained in such a way that let the reader build a stepwise story of the role of calpains in atherosclerosis. It is suggested to use letters or numbers in the figure to guide the reading and interpretation of this figure. Moreover, even when the figures were created and integrated (not previously existed in this presentation) the author should give credit to the publications used to get this result. There are abbreviations in the figure not explained, such as: Rac1, eIF4A3, IL-10. The before mentioned arguments are in line to provide a self-explanatory message with tables or figures and this recommendation can be extended to the other figures in the manuscript. 

Line 72, a there is a mistake with the word “dcommon”

The information about CAPN10 is poorly described. This section on the results of diabetes and its variants (gestational, type II, etc.) can be improved, showing a deeper explanation of the functional impact of SNP43 on the CAPN10 gene. It suddenly popped out that SNP-43 plays a role in insulin secretion but is not clearly stated.

There is amazing work in this review; there is no doubt about the considered information and its integration through different experiments and outcomes that position calpains as a possible and interesting molecular target for cardiovascular traits. There are some points for improvement before it is published.

Author Response

[Author] Thank you so much for your valuable comments and suggestions. Here I will address your question and suggestion in a point-to-point manner.

The article titled “Calpain and cardiometabolic diseases” was reviewed.

The aim of the review was to point out the role of calpains in the complications of cardiometabolic diseases and molecular targets.

The author starts briefly describing different calpains in terms of their structure and homology. Subsequently, there is information over the molecular basis of cardiometabolic calpain isozymes and calpastatin. Finally, the author structured the manuscript in different sections over calpain and atherosclerosis, diabetes, liver disease and obesity.

The author has a broad number of publications on Calpain function, regulation, and isoforms. Which they are mostly presented in the review. 

Comments and suggestions

It is clearly understood this is not a Systematic review. However, it is suggested to give an overview of the researching process to gather and filtering the information. If it Is the purpose to present all the information since its discovery and relation with cardiometabolic traits or any criteria to include the articles (outcomes, methodology such as in vivo and in vitro models, etc.).

It should be specified the calpains included in the manuscript are 1, 2, 6 and 10 because of their involvement in the cardiometabolic traits and consequences, and they are presented as conventional and unconventional calpains.

Please check the connector “and” in the following idea, lines 292-294:

[Author] The text was corrected as follows.

––––Moreover, the protein expression of Rac1 in bone marrow-derived macrophages exhibited a negative correlation with that of calpain-6  (lines 310-312 in the revised manuscript)

The figures (graphical material) give a concise message together with the written information. In this sense, it is necessary to structure the figures to clarify the idea. In figure legend 2, it is explained in such a way that let the reader build a stepwise story of the role of calpains in atherosclerosis. It is suggested to use letters or numbers in the figure to guide the reading and interpretation of this figure. Moreover, even when the figures were created and integrated (not previously existed in this presentation) the author should give credit to the publications used to get this result. There are abbreviations in the figure not explained, such as: Rac1, eIF4A3, IL-10. The before mentioned arguments are in line to provide a self-explanatory message with tables or figures and this recommendation can be extended to the other figures in the manuscript.

[Author] Each material in Figures 2-4 was labeled by letters. Accordingly, related figure legends were modified. Abbreviations were fully expanded in figure legends.

Line 72, a there is a mistake with the word “dcommon”

[Author] The word “dcommon” was replaced with “the common”.

The information about CAPN10 is poorly described. This section on the results of diabetes and its variants (gestational, type II, etc.) can be improved, showing a deeper explanation of the functional impact of SNP43 on the CAPN10 gene. It suddenly popped out that SNP-43 plays a role in insulin secretion but is not clearly stated.

[Author] The explanation of SNP43 on the CAPN10 gene was noted as follows.

––––Conventional calpains have been reported to be activated in hyperglycemia to blunt endothelial cells, hepatocytes, and pancreatic β cells (Figure 3). Around the initial years of the 21st century, polymorphism of the CAPN10 (calpain-10 gene) has been reported to increase the risks of type 2 diabetes and decrease glucose disposal. Indeed, the G allele of the single nucleotide polymorphism (SNP)-43 has shown associations with insulin resistance [80] and the onset of type 2 diabetes in both cross-sectional [81] and prospective studies [82]. While other investigations have failed to establish a correlation between the G allele and type 2 diabetes or insulin resistance [83–85], additional SNPs and/or haplotypes within the corresponding locus have been identified to possess greater significance in glucose metabolism [83,86,87]. Furthermore, variations in this genetic locus might govern insulin secretion [88]. In addition, recent studies have identified that hyperglycemia-induced conventional calpains contribute to myocyte injury in the ischemic heart. (lines 371-380, 396-397 in the revised manuscript)

There is amazing work in this review; there is no doubt about the considered information and its integration through different experiments and outcomes that position calpains as a possible and interesting molecular target for cardiovascular traits. There are some points for improvement before it is published.

Reviewer 3 Report

Comments and Suggestions for Authors

This is an excellent review of Calpain and cardiometabolic diseases. The insights provided allow readers to develop a better understanding of the effects of this stress-responsive intracellular protease and its role in the pathogenesis of cardiometabolic disorders, such as atherosclerosis, diabetes, and hepatic disease. I enjoyed reading this article and congratulate you on a great effort.

You have mentioned some influences from drugs and diets on the disease progression and calpains. I was wondering if it was possible to include some information in the review on the role of exercise and its interaction with calpains.

I noticed a few minor edits that could enhance the clarity of the content:

Line 71: Add a full stop after "cardiometabolic diseases."

Line 293: Revise the sentence for better readability, such as: "...analyzed the expression of RhoGTPase and observed a noticeable negative correlation between calpain-6 and Rac1 protein expression."

Line 365: Please define "SNP-43" for clarity.

Additionally, there are inconsistent font sizes and styles throughout the manuscript. Kindly ensure consistency in formatting.

Your consideration of including information about the role of exercise and its interaction with calpains would further enrich the review.

Comments on the Quality of English Language

Excellent quality of English overall. 

Line 71: Add a full stop after "cardiometabolic diseases."

Line 293: Revise the sentence for better readability, such as: "...analyzed the expression of RhoGTPase and observed a noticeable negative correlation between calpain-6 and Rac1 protein expression."

Author Response

[Author] Thank you so much for your valuable comments and suggestions. Here I will address your question and suggestion in a point-to-point manner.

This is an excellent review of Calpain and cardiometabolic diseases. The insights provided allow readers to develop a better understanding of the effects of this stress-responsive intracellular protease and its role in the pathogenesis of cardiometabolic disorders, such as atherosclerosis, diabetes, and hepatic disease. I enjoyed reading this article and congratulate you on a great effort.

You have mentioned some influences from drugs and diets on the disease progression and calpains. I was wondering if it was possible to include some information in the review on the role of exercise and its interaction with calpains.

I noticed a few minor edits that could enhance the clarity of the content:

Line 71: Add a full stop after "cardiometabolic diseases."

[Author] The period was inserted.

Line 293: Revise the sentence for better readability, such as: "...analyzed the expression of RhoGTPase and observed a noticeable negative correlation between calpain-6 and Rac1 protein expression."

[Author] The text was corrected as follows.

––––Moreover, the protein expression of Rac1 in bone marrow-derived macrophages exhibited a negative correlation with that of calpain-6  (lines 310-312 in the revised manuscript)

Line 365: Please define "SNP-43" for clarity.

[Author] The explanation of SNP43 on the CAPN10 gene was noted as follows.

––––Conventional calpains have been reported to be activated in hyperglycemia to blunt endothelial cells, hepatocytes, and pancreatic β cells (Figure 3). Around the initial years of the 21st century, polymorphism of the CAPN10 (calpain-10 gene) has been reported to increase the risks of type 2 diabetes and decrease glucose disposal. Indeed, the G allele of the single nucleotide polymorphism (SNP)-43 has shown associations with insulin resistance [80] and the onset of type 2 diabetes in both cross-sectional [81] and prospective studies [82]. While other investigations have failed to establish a correlation between the G allele and type 2 diabetes or insulin resistance [83–85], additional SNPs and/or haplotypes within the corresponding locus have been identified to possess greater significance in glucose metabolism [83,86,87]. Furthermore, variations in this genetic locus might govern insulin secretion [88]. In addition, recent studies have identified that hyperglycemia-induced conventional calpains contribute to myocyte injury in the ischemic heart. (lines 371-380, 396-397 in the revised manuscript)

Additionally, there are inconsistent font sizes and styles throughout the manuscript. Kindly ensure consistency in formatting.

[Author] The Format of the manuscript was confirmed. 

Your consideration of including information about the role of exercise and its interaction with calpains would further enrich the review.

Round 2

Reviewer 1 Report

Comments and Suggestions for Authors

 The revised manuscript was well-designed and the paper was well written. 

Comments on the Quality of English Language

Minor editing of English language required